# Boosting 3D Object Detection with Density-Aware Semantics-Augmented Set Abstraction

**DOI:** 10.3390/s23125757

**Published:** 2023-06-20

**Authors:** Tingyu Zhang, Jian Wang, Xinyu Yang

**Affiliations:** 1College of Computer Science and Technology, Jilin University, Changchun 130012, China; zhangty21@mails.jlu.edu.cn; 2Key Laboratory of Symbolic Computation and Knowledge Engineering of Ministry of Education, Jilin University, Changchun 130012, China; 3China Automotive Innovation Corporation, Nanjing 210000, China; yangxy18@mails.jlu.edu.cn

**Keywords:** LiDAR, 3D object detection, autonomous driving, set abstraction, farthest point sampling

## Abstract

In recent years, point cloud-based 3D object detection has seen tremendous success. Previous point-based methods use Set Abstraction (SA) to sample the key points and abstract their features, which did not fully take density variation into consideration in point sampling and feature extraction. The SA module can be split into three parts: point sampling, grouping and feature extraction. Previous sampling methods focus more on distances among points in Euclidean space or feature space, ignoring the point density, thus making it more likely to sample points in Ground Truth (GT) containing dense points. Furthermore, the feature extraction module takes the relative coordinates and point features as input, while raw point coordinates can represent more informative attributes, i.e., point density and direction angle. So, this paper proposes Density-aware Semantics-Augmented Set Abstraction (DSASA) for solving the above two issues, which takes a deep look at the point density in the sampling process and enhances point features using onefold raw point coordinates. We conduct the experiments on the KITTI dataset and verify the superiority of DSASA.

## 1. Introduction

Due to its numerous applications in fields such as robotics, virtual reality, and autonomous vehicles, 3D object detection has drawn significant attention. LiDAR sensors have been broadly employed in autonomous driving systems, which can capture the environment surrounding the host vehicle. Compared to the camera, LiDAR can obtain precise 3D contours of objects, thereby enhancing the performance of 3D object detection.

Point cloud-based 3D object detection methods can be roughly classified into three categories, namely point-based, voxel-based and hybrid-based detectors. Voxel-based methods discretize points to regular grids and use sparse 3D convolution [1,2] to extract the voxel feature. However, voxel-based methods cannot avoid quantization loss because of voxelization. Point-based methods benefit from the pioneer PointNet series methods [3,4,5], which directly operate on raw point clouds to obtain the point-level features. Hybrid-based methods [6,7] fuse the aforementioned two means, making full use of the efficiency of voxel-based methods and the highly accurate contextual information extracted by point-based methods.

This paper only focuses on the point-based methods which do not introduce the quantization loss. Multiple point-based methods [8,9,10] use PointNet++ [4] and its variants as their backbone, where the SA module is the most important. The SA module can be divided into three steps: sampling, grouping and feature extraction. The Farthest Point Sampling (FPS) [4] is commonly used in the sampling process, aiming to sample key points evenly distributed throughout the entire point clouds. Nevertheless, the FPS only considers the distribution balance of the sampling points; thus, it cannot ensure that the sampling points are related to the objects. Previous methods tend to sample more foreground points to increase the recall rate [9,10,11]. Despite the considerable success of these sampling techniques, they cannot avoid the issue of the sampling density imbalance among the foreground objects. For example, they may sample more points in objects with dense points and sample fewer points in objects with sparse points. This leads to sampling variances among foreground points with various density. It is worth noting that objects with dense points often involve sufficient contextual information, so it is unnecessary to sample more points compared to distant or occluded objects, which may have much fewer points. This problem can be alleviated by combining the distance in Euclidean space and feature space. However, this paper argues that the explicit inclusion of density variance is more straightforward. In this paper, we propose a new sampling strategy called Density-Semantics-aware Farthest Point Sampling (DS-FPS), which takes the point-level confidence score and density into account.

In previous work, the SA module primarily focuses on the high-level features of points, only encoding the low-level relative coordinates. However, raw coordinates of the points contain valuable information that expresses spatial position relations. Therefore, we propose the Raw Coordinate Enhancement (RCE) module to further capture the local context with minimal increase in computing resource. To sum up, our contributions are as follows:We propose the DSASA framework, which includes the DS-FPS and the RCE module to balance the foreground points sampling and enhance the point features.We conduct experiments to verify that the DS-FPS can alleviate sampling imbalance, and the RCE module can improve performance with negligible increases in computing resources.The evaluation conducted on KITTI [12] 3D benchmarks shows that DSASA outperforms other single-staged point-based detectors under the same experimental environment in the outdoor scenarios.

## 2. Related Work

### 2.1. Point Cloud-Based 3D Detectors

Point cloud-based 3D object detection can be loosely divided into three groups, such as voxel-based detectors, point-based detectors and hybrid-based detectors.


**Voxel-based 3D Detectors**


Voxel-based 3D detectors [2,13,14] discretize 3D space into regular 3D grids, which can be processed by 3D convolution. VoxelNet [13] is the pioneer work that applies 3D convolution to the point cloud in an end-to-end fashion. However, due to the sparsity of the point cloud, common 3D convolution may mainly operate on the empty 3D grid, wasting the computing resources. SECOND [2] applies sparse 3D convolution and submanifold 3D convolution [1] to the 3D object detection, utilizing the GPU hash table to accelerate the training and inference process. To further reduce the inference time, PointPillars [14] compress the height dimension to create the pseudo-image feature map, making it possible to apply efficient 2D convolution. Votr [15], VoxSeT [16] and CT3D [17] introduce the creative Transformer [18] to voxel-based methods for enhancing the feature interaction. SST [19] proposes the region grouping for region attention which avoids the widely used down-sampling operation in voxel-based methods, thus maintaining large receptive fields all the way.


**Point-based 3D Detectors**


Another stream is the point-based 3D detectors which directly operate on a raw point cloud. Profiting from the pioneer PointNet series methods [3,4,5], point-based methods rely on the SA module to down-sample point clouds and abstract down-sampled point features. PointRCNN [8] uses the SA module to down-sample the points and obtain their semantic features; then, it utilizes the Feature Propagation (FP) layer to propagate the subset features to the universe set. With the semantic features of points as input, the detection head generates predictions in the canonical coordinates system. The FPS in the vanilla SA module only takes into account the points distribution, whereas it cannot guarantee the sampled points to be beneficial to the prediction generation. Hence, Yang et al. [9] propose 3DSSD, which replaces the partial vanilla FPS with the novel Feature-based FPS (F-FPS), which conforms to the assumption that features of background points are similar and features of foreground points in different instances vary a lot. Using 3DSSD implicitly leads the model to sampling more foreground points, while SASA [10] and IA-SSD [11] explicitly use a foreground prediction module, guiding the model to sample more foreground points, which further increases the instance recall. In addition to the PointNet series methods, Graph Neural Network (GNN) is also an alternative in point-based methods. Point-GNN [20] constructs the graph on the voxel-down-sampled point cloud and employs GNN on the local neighborhood to iteratively update the vertex features. BADet [21] does not construct the graph on raw point clouds, but it focuses on the high-level Region-of-Interest (RoI) semantic features. Due to relatively slow FPS, point-based detectors take longer to infer, yet a direct operation on a raw point cloud preserves the point cloud structure.


**Hybrid-based 3D Detectors**


It is straightforward to mutually conduct transformation between voxel features and point features. For instance, voxel features can be obtained through aggregation operations (e.g., max-pooling, average-pooling) on the points in each voxel. Point features can be obtained via the interpolation of neighboring voxel features. Many researchers fuse voxel-based and point-based methods into a single architecture, which makes full use of the efficiency of voxel-based methods and the precise local structure maintenance of point-based methods. PV-RCNN [6] uses FPS to sample key points, and it aggregates point features, voxel features and Bird’s Eye View (BEV) features to form more representative key point features in the first stage. In the second stage, proposals are split into regular grids. With the help of a small quantity of key points, grid points can capture wider receptive fields, thus helping refine the proposal. M3DETR [22] uses the Transformer [18] to model the correlation among different types of features, which varies from the simple concatenation in PV-RCNN [6]. PV-RCNN++ [7] further narrows down the searching space, making it more efficient. The hybrid-based methods take advantages of both voxel-based and point-based methods for 3D point-cloud feature learning while at the expense of computation.

The proposed DSASA is a point-based 3D object detector, as point-based methods are capable of preserving more geometric features and achieving a better balance between performance and efficiency when compared to voxel-based methods and hybrid-based methods.

### 2.2. Point Sampling in Point Cloud Processing

The FPS in PointNet++ [4] is the most commonly used method in point cloud down-sampling. It iteratively samples the point which has the largest distance with the already sampled points set. The vanilla FPS focuses on the distance in the Euclidean space, which cannot guarantee the instance recall. However, not all points are equal in point clouds  [11], even distribution is not optimal for point cloud processing, leading to more sampling methods coming into being. It is important to note that 3DSSD [9] assumes that background points share similar features and foreground point features differ from each other. Based on this assumption, 3DSSD proposes F-FPS, which tends to have a better cover of foreground points. Merely sampling foreground points is not conducive to the classification due to the imbalance between foreground and background points, so SSD samples half of the points using FPS and samples the rest using F-FPS to balance the amount. The method employed in 3DSSD is implicit, while SASA [10] and IA-SSD use more forthright means that use Semantic-aware FPS (S-FPS) and simply sample points with a top K confidence score to make the model explicitly collect more foreground points.

However, the above sampling methods do not take into account the density variances among instances, thus leading to near instances which have enough information to sampling more points and distant instances which have insufficient information to sampling less points. To alleviate this issue, we encode the point density as the weight of a distance metric, namely DS-FPS. The comparisons of FPS, F-FPS, S-FPS and DS-FPS are depicted in Figure 1.

### 2.3. Point Density in 3D Object Detection

One of the reasons that a point cloud cannot be processed as an image is the density variances, so it is significant to take point density into consideration. Previous studies in this area have been inadequate. PDV [23] is a pioneer 3D detector which encodes the point density into the point feature. There are two alternatives to encode the point density in PDV: one is Kernel Density Estimation (KDE) [24,25], and the other is simply using the logarithm of the points amount in the neighborhood. Pyramid-RCNN proposed the Density-Aware Radius Prediction module, which uses the point density to determine the focusing range of RoIs dynamically.

Considering the limited research on density-based methods and the significance of the density attribute, we integrate the density attribute into the FPS and the feature extraction process, resulting in improved performance.

### 2.4. Learning from Raw Points Coordinates

In the feature extraction stage of the SA module, previous work [8,9,10] concentrates on high-level semantic features. Raw point coordinates are only used in the model input and the calculation of relative coordinates. However, coordinates are the original feature of points which contain rich information such as direction angle and density. PDV [23] utilizes KDE [24,25] to encode the point density of the local region, which depends on the coordinates and concatenates it with high-level semantic features for the next module input. PointPillars [14] encodes the pillar center and relative coordinates by the raw points position.

To enhance the utilization of raw point coordinates, we introduce the RCE module, which takes only raw point coordinates as input and transforms them into more informative attributes. This leads to improved performance without imposing significant computing resource requirements.

## 3. Methods

In this section, we first overview the vanilla SA module in Section 3.1. Then, we introduce the architecture of DSASA in Section 3.2 and describe DS-FPS and RCE in detail.

### 3.1. Preliminary

The vanilla SA module can be split into three parts: (i) sampling, (ii) grouping and (iii) feature extraction, which is shown in Figure 2.

#### 3.1.1. Sampling

FPS is the most commonly used sampling method, which guarantees the sampling points are evenly distributed in 3D space. Many researchers refine the sampling strategy for more reasonable points distribution. FPS and its variants can be generalized as Algorithm 1. Previous sampling methods only vary in the Sample function, dist array and Update function. We take Distance-based FPS (D-FPS), namely vanilla FPS [4], F-FPS [9] and S-FPS  [10], as examples and compare them from the above three perspectives.

**D-FPS** The Sampling function in Point-RCNN [8] and PointNet++ [4] involves randomly selecting a point in the point cloud, which is often the first one saved in data in practice. The dist array can be calculated by $dist_{k}=d_{k}$, where $d_{k}$ is the *k*th point’s minimal distance with the sampled set in Euclidean space. The Update function can be denoted as Equation (Equation 1)
(1)$$d_{j}=min(d_{j},∥x_{j}-x_{k_{i}}{∥}_{2}),j\in \left\{1,\ldots ,N\right\}$$
where $k_{i}$ is the index of the sampled point in this iteration, *N* is the total point number in this iteration, and ${∥\cdot ∥}_{2}$ means the Euclidean norm.

**F-FPS** The Sampling function and dist array in F-FPS [9] are the same as in D-FPS. The only changed Update function can be formulated as Equation (Equation 2)
(2)$$d_{j}=min(d_{j},\mu ∥x_{j}-x_{k_{i}}{∥}_{2}+∥f_{j}-f_{k_{i}}{∥}_{2}),j\in \left\{1,\ldots ,N\right\}$$
where μ is the balance factor to balance the feature distance and the coordinate distance.

**S-FPS** SASA [10] conducts point segmentation to encourage the model to sample more foreground points. The first sampled points can be determined by confidence scores rather than random sampling. The Sampling method in S-FPS can be determined by argmax function as illustrated in Equation (Equation 3).
(3)Sampling(Input)=argmax(S)
where *S* is the confidence score set of the input points. The dist array can be formulated as the confidence weighted distance as illustrated in Equation (Equation 4).
(4)$$dist_{k}=p_{k}^{\gamma }*d_{k}$$
where $p_{k}$ is the *k*th point confidence score and γ is the balance factor. The Update function is consistent with D-FPS.
**Algorithm 1: ** Generalized Farthest Point Sampling**Input: **
 (required):
 coordinates $X=\left\{x_{1},\ldots ,x_{n}\right\}\in R^{N\times 3}$
 (optional):
 features $F=\left\{f_{1},\ldots ,f_{n}\right\}\in R^{N\times d}$
 foreground scores $S=\left\{s_{1},\ldots ,s_{n}\right\}\in R^{N\times 1}$
**Output: **
 sampled key point set $K=\left\{k_{1},\ldots ,k_{n}\right\}\in R^{M\times 3}$
1: initialize an empty sampling point set *K*;
2: initialize a distance array *d* of length *N* with all +∞;
3: initialize a visit array *v* of length *N* with all zeros;
4: **for**
i=1 to *M* **do**
5:  **if** i=1 **then**
6:   $k_{i}=Sample\left(Input\right)$
7:  **else**
8:   $D=\{dist_{k}|v_{k}=0\}$
9:   $k_{i}=\arg\max\left(D\right)$
10:  **end if**
11:  add $k_{i}$ to *K*, $v_{k_{i}}=1$
12:  **for**
j=1 to *N* **do**
13:   $Update(d_{j})$
14:  **end for**
15: **end for**


#### 3.1.2. Grouping

Due to the maldistribution of the point cloud, researchers often use a ball query to group the neighboring points rather than K-Nearest Neighborhood (KNN). To obtain multi-scale features, previous work [8,9,10] uses a different ball radius to group points and aggregates features through concatenation. In the SASA [10] source code and MMDetection3d repository [26], they use a dilated ball query to group the point features. The input of the grouping module includes the coordinates of current stage points $P=\left\{p_{1},\ldots ,p_{N}\right\}\in R^{N\times 3}$, the features of current stage points $F=\left\{f_{1},\ldots ,f_{N}\right\}\in R^{N\times d}$ and the coordinates of sampled points $C=\left\{c_{1},\ldots ,c_{N}\right\}\in R^{M\times 3}$. We therefore compare the vanilla ball query and the dilated ball query.

**Vanilla Ball Query** The first step of the ball query is to determine the grouping points index of sampled points, which can be formulated as below.
(5)$${g\_idxs}_{k}^{i}=\{j|∥p_{j}-c_{i}∥⩽radius_{k},j=1,\ldots ,N,i=1,\ldots ,M,k=1,\ldots ,K\}\in R^{nsample}$$
where *N* and *M* are the input points number and sampled points number of this SA module, $radius_{k}$ is the *k*th ball query radius, and $c_{i}$ is the *i*th sampled point coordinate. $p_{j}$ is the *j*th input points coordinate. nsample denotes the number of neighboring points required to be grouped. If the number of neighboring points is fewer than nample, we pad by repeating the existing points. Otherwise, if the number of neighboring points is more than nample, we random sample nsample points.

**Dilated Ball Query** The sole difference between the vanilla and dilated ball query is the grouping radius. In the dilated ball query, the grouping ranges have no intersection; the *k*th grouped points indexes can be formulated as Equation (Equation 6)
(6)$$\begin{array}{c} {g\_idxs}_{k}^{i}=\{j|radius_{k-1}⩽∥p_{j}-c_{i}∥⩽radius_{k},j=1,\ldots ,N,i=1,\ldots ,M,k=1,\ldots ,K\}\in R^{nsample} \end{array}$$

The variables are identical to Equation (Equation 5), and it is worth noting that $radius_{0}$ is set to zero as default. Once we obtain the *k*th group points index, the following steps are consistent in both ball query methods. We can group the points features to obtain the *i*th ball group feature in the *k*th level as demonstrated in Equation (Equation 7).
(7)$$\begin{array}{cc} {g\_feature}_{k}^{i}= & \left\{Concat\left(f_{j}\right)|j\in g\_idxs_{k}^{i}\right\}\in R^{nsample\times d} \end{array}$$
where *d* is the dimension of input point features, and Concat means the concatenation operation. Afterwards, we can use concatenation to obtain the multi-scale sampled point features as demonstrated in Equation (Equation 8).
(8)$$s^{i}=\{Concat({g\_feature}_{k}^{i})|k=1,\ldots ,K\}\in R^{nsample\times (kd)}$$

The sampled point features will be further fed to the feature extraction module.

#### 3.1.3. Feature Extraction

To further extract point features, it is common practice to use the Multi-Layer Perceptron (MLP) to capture more refined sampled point features and use the pooling operation to aggregate the sampled point features, which is formulated as below.
(9)$$\begin{array}{c} s^{i}=Pooling\left(MLP\left(s^{i}\right)\right)\in R^{D} \end{array}$$
where *D* is the dimension of the output point features.

### 3.2. Density-Aware Semantics-Augmented Set Abstraction

We make two main modifications to the vanilla SA module, that is the DS-FPS and the RCE module, and we follow SASA to use the dilated ball query. The overall architecture is depicted in Figure 3. We will present the above two modules in Section 3.2.1 and Section 3.2.2

#### 3.2.1. Density-Aware Semantic Farthest Point Sampling

To encode point density in FPS, there are two issues need to be handled: first, how to represent point density; secondly, how to explicitly add point density to FPS.

**How to represent point density?** As described in Section 2.3, KDE and a simple logarithm function can be used to represent the point density. We choose the later for simplicity. Using the logarithm function, point density can be represented as below.
(10)$$density=\left\{\begin{array}{cc} \; & log(\sum_{i=1}^{K}count_{i}),ifdilated\\ \; & log(count_{K}),otherwise \end{array}\right.$$
where log is the base-10 logarithm function, and $count_{i}$ means the point number in the *i*th query space. *K* is the amount of query space. ifdilated means if the type of ball query type is a dilated ball query.

**How to add point density to FPS?** As is described in Section 3.1.1, sampling methods differ in the Sample function, dist array and Update function. We keep the Sample function and Update function the same as SASA [10]. For the dist array, we expect the points with low density to have farther distance, so we utilize the negative value of the sigmoid function to encode the weight, reflecting the inverse relationship between density and distance.
(11)$$dist_{k}=p_{k}^{\gamma }*d_{k}*{\left(1-sigmoid\left(density\right)\right)}^{\lambda }$$
where sigmoid is the Sigmoid function, and density is the same as that defined in Equation (Equation 10). In Equation (Equation 11), we plus one to let the density weight lie between 0 and 1. γ and λ are the balance factors to balance the confidence weight and the density weight.

#### 3.2.2. Raw Coordinate Enhancement

There are many useful attributes that need to be discovered based on points coordinates. We denote the ball center as $\left(x_{1},y_{1},z_{1}\right)$ and one of the neighboring points in the ball as $\left(x_{2},y_{2},z_{2}\right)$ in this section.

**Relative position in the query ball** Inspired by the notion of proposal ambiguity put forward in LiDAR-RCNN [27], we posit that the relative position within the query ball is crucial in providing the model with additional information about the local context. As illustrated in Figure 4, it is imperative to guide the model in discerning the grouping boundaries effectively. The method of using the normalized relative position in the ball query is effective. Therefore, we encode the relative position within the query ball as described in Equations (Equation 12) and (Equation 13).
(12)$$r_{offset}=r_{out}-r_{in}$$
(13)$$rel\_pos=(\left(x_{2}-x_{1}-r_{in}\right)/r_{offset},\left(y_{2}-y_{1}-r_{in}\right)/r_{offset},\left(z_{2}-z_{1}-r_{in}\right)/r_{offset})$$
where $r_{in}$ and $r_{out}$ are the smaller and the larger query radius in dilated ball query, respectively.

**Relative direction angle** The relative direction angle of the neighboring points can be encoded by the relative coordinates as below.
(14)$$dist_{1}=\sqrt{{\left(x_{2}-x_{1}\right)}^{2}+{\left(y_{2}-y_{1}\right)}^{2}}$$
(15)$$dist_{2}=\sqrt{{\left(y_{2}-y_{1}\right)}^{2}+{\left(z_{2}-z_{1}\right)}^{2}}$$
(16)$$dist_{3}=\sqrt{{\left(z_{2}-z_{1}\right)}^{2}+{\left(x_{2}-x_{1}\right)}^{2}}$$
(17)$$\theta_{1}=atan2\left(z_{2}-z_{1},dist1\right)$$
(18)$$\theta_{2}=atan2\left(x_{2}-x_{1},dist2\right)$$
(19)$$\theta_{3}=atan2\left(y_{2}-y_{1},dist3\right)$$
(20)$$dir_{rel}=\left(sin\left(\theta_{1}\right),cos\left(\theta_{1}\right),sin\left(\theta_{2}\right),cos\left(\theta_{2}\right),sin\left(\theta_{3}\right),cos\left(\theta_{3}\right)\right)$$
where atan2 is the inverse tangent function. The illustration of $\theta_{1}$ is depicted in Figure 5. Although relative direction angles are implicitly contained in relative coordinates, we argue that the explicitly encoding is a more reasonable approach, which helps the network focus on the direction relation.

**Density** Inspired by PDV [23], we encode the logarithm amount of neighboring points as density, which helps network realize the distribution of the neighborhood.

Combined with the above three attributes, we add 10 channels to enhance the point features.

## 4. Experiments

We call our model DSASA. DSASA is evaluated on the challenging 3D object detection benchmark of the KITTI dataset.

### 4.1. Datasets

The KITTI dataset is a widely used benchmark in 3D object detection. It contains 7481 LiDAR point clouds as well as finely calibrated 3D bounding boxes for training and 7518 samples for testing. Following SECOND [2], we split training samples into a training set with 3712 samples and a validation set with 3769 samples; then, we use this partition to find the optimal hyper-parameters. To obtain the final results, which need to be submitted to the KITTI test server, we followed PV-RCNN [6] where 80% of the training samples are used for training, and the remaining 20% are used for validation.

### 4.2. Implementation Details

Most of the architecture is the same as SASA [10]. We replace SASA with our proposed DSASA. It is worth noting that SASA [10] trains on four GPUs with a batchsize of four per GPU. However, due to the limited training resources, we train with a batchsize of eight on a single RTX4090. The learning rate and other hyper-parameters are the same as SASA. We set λ in Equation (Equation 11) to 1.0. The reason for choosing 1.0 is detailed in Section 4.4.

### 4.3. Main Results

Table 1 presents the performance of 3D object detection specifically for the Car class on the KITTI test server. Due to the limited GPU resources and the random testing set partition, we cannot fully reproduce the results mentioned in SASA [10]. So we keep other training configuration the same as SASA except for batchsize and testing set partition, then set the model trained with batchsize 8 on single RTX4090 as the baseline, and DSASA is better than the baseline and other single stage point-based methods. The qualitative results are depicted in Figure 6.

### 4.4. Ablation Study

**Car detection performance on validation set** As presented in Table 2, we choose the established outdoor 3D detectors, 3DSSD and PointRCNN, which already contain the SA module, as our baselines. Independently, we incorporate SASA and our proposed method, DSASA, with these baselines and evaluate their performance on the validation set. Our methods demonstrate superior performance compared to the baselines, even when combined with SASA, under the same experimental conditions. Figure 7 showcases the qualitative results.

**Multi-classes detection performance on validation set** As shown in Table 3, we conduct a similar experimental setup as in Table 2, with the exception that the models classify three classes. In the single-class detection model, we directly predict the dimensions of the instances. In the multi-class detection model, we modify the detection head to classify three classes and predict the dimension offset between the predictions and the mean size of each class in the KITTI dataset. The qualitative results are depicted in Figure 8. As demonstrated in Table 3, DSASA achieves improved performance especially in detecting small objects. This can be attributed to the fact that small objects typically exhibit lower density, and DSASA effectively addresses this by sampling more points within such objects.

**Effects of density balance factor** We compare DS-FPS with different balance factors λ in Figure 9. The extremely small or large number will interrupt the final results, so we set λ to be 1.0.

**Effects of Different attributes encoded in RCE** We set DSASA without the RCE module as our baseline and compare the performance using various attributes in Table 4. We not only conduct experiments on the three attributes mentioned in Section 3.2.2 but also take the Absolute Direction Angle (ADA) into account, which can be formulated as below, where RPQB means Relative Position in Query Ball, RDA means the Relative Direction Angle, Density means the point density and ADA means the Absolute Direction Angle.
(21)$$abs\_dist_{1}=\sqrt{x_{2}^{2}+y_{2}^{2}}$$
(22)$$abs\_dist_{2}=\sqrt{y_{2}^{2}+z_{2}^{2}}$$
(23)$$abs\_dist_{3}=\sqrt{z_{2}^{2}+x_{2}^{2}}$$
(24)$${abs\_\theta }_{1}=atan2\left(z2,abs\_dist1\right)$$
(25)$${abs\_\theta }_{2}=atan2\left(x2,abs\_dist2\right)$$
(26)$${abs\_\theta }_{3}=atan2\left(y2,abs\_dist3\right)$$
(27)$$dir_{abs}=\left(sin\left({abs\_\theta }_{1}\right),cos\left({abs\_\theta }_{1}\right),sin\left({abs\_\theta }_{2}\right),cos\left({abs\_\theta }_{2}\right),sin\left({abs\_\theta }_{3}\right),cos\left({abs\_\theta }_{3}\right)\right)$$

First, we add a single attribute to the RCE module to test its validity. Next, we combine three attributes together to examine their collective effectiveness. We intentionally exclude the combination of ADA and RDA, since ADA and RDA both pertain to direction angles, and we aim to avoid redundant attributes. Simply adding a single attribute can boost the performance, and combining RPQB, RDA and Density boosts the most. We think that the Absolute Direction Angle is similar among points in the same bounding box, so it is not as distinctive as the other three attributes.

**Verify the validation of RCE** We have a doubt as to whether the attributes in RCE are really useful or the boost is due to the more learnable parameters introduced by MLP. So, we conduct the experiments with two strategies. One is that we insert linear layer, BN layer and ReLU in the beginning of feature extraction stage, which learns 10 channels (six in RDA, three in RPQB and one in Density) from coordinates. Then, we concatenate the generated features with input features to form features with C+10 channels, where *C* is the channel number of the input features, which is fed into the following MLPs. Another strategy adds more learnable parameters to the model. It utilizes an MLP to convert the channels from C+3 to C+13 and sends the output features to the following modules. However, these two strategies did not boost more than the RCE module in Table 5, so we are convinced that the performance increase has little relation with the extra learnable parameters but rather affects the meticulous design. We denote the first strategy as Small MLP and the second as Large MLP.

**Sampling means and variances** The proposed DS-FPS aims to balance the sampling process among multiple instances, so we study the average point number sampled in GT and the standard deviation (Std) among the foreground point amount. The mean and Std can be calculated as below.
(28)$$\begin{array}{c} Mean=(\sum_{i=0}^{N}cnt_{i})/N \end{array}$$
(29)$$\begin{array}{c} Std=sqrt(\sum_{i=0}^{N}{\left(cnt_{i}-Mean\right)}^{2}) \end{array}$$
where *N* is the total GT number, and $cnt_{i}$ is the point number in the *i*th GT. As described in Table 6, the DS-FPS samples more points than F-FPS and shows less variance than S-FPS.

**Computing burden introduced by DSASA** As shown in Table 2, the PointRCNN+DSASA takes 2 ms more than PointRCNN+SASA, and 3DSSD+DSASA takes 1 ms more than 3DSSD+SASA. We assume that 1 ms and 2 ms are negligible in detection, since the LiDAR frequency is often 20 Hz, so we verify that DSASA can boost the detection performance with little cost.

## 5. Conclusions

In this article, we propose DSASA. The previous SA module either takes more attention to the even point sampling or purges the model to sample more foreground points. DSASA considers both point density and confidence scores, aiming to achieve a more balanced sampling process. In the second SA module, DS-FPS in DSASA samples 94% more foreground points than F-FPS, and the Std in the sampling process is reduced by 30% compared to S-FPS. Furthermore, the proposed RCE module in DSASA utilizes raw coordinates to extract valuable information, resulting in improved performance with only a 1 ms increase in inference time.

However, the proposed DS-FPS is based on FPS series methods, which have a time complexity of $O(n^{2})$ and are not efficient for large-scale point clouds. On the other hand, simply choosing points with top K foreground scores can provide faster processing speed, but it relies heavily on foreground segmentation performance. In the future, it is worth studying how to strike a balance between performance and efficiency in sampling methods. Additionally, although the cascaded ball query expands the receptive field, its range is still limited, so using Transformer to obtain the global receptive fields is a better choice. We only use a single dataset for verification, which makes it less convictive. We will conduct more experiments on diverse datasets to demonstrate the feasibility in our following work.

## Figures and Tables

**Figure 1 sensors-23-05757-f001:**
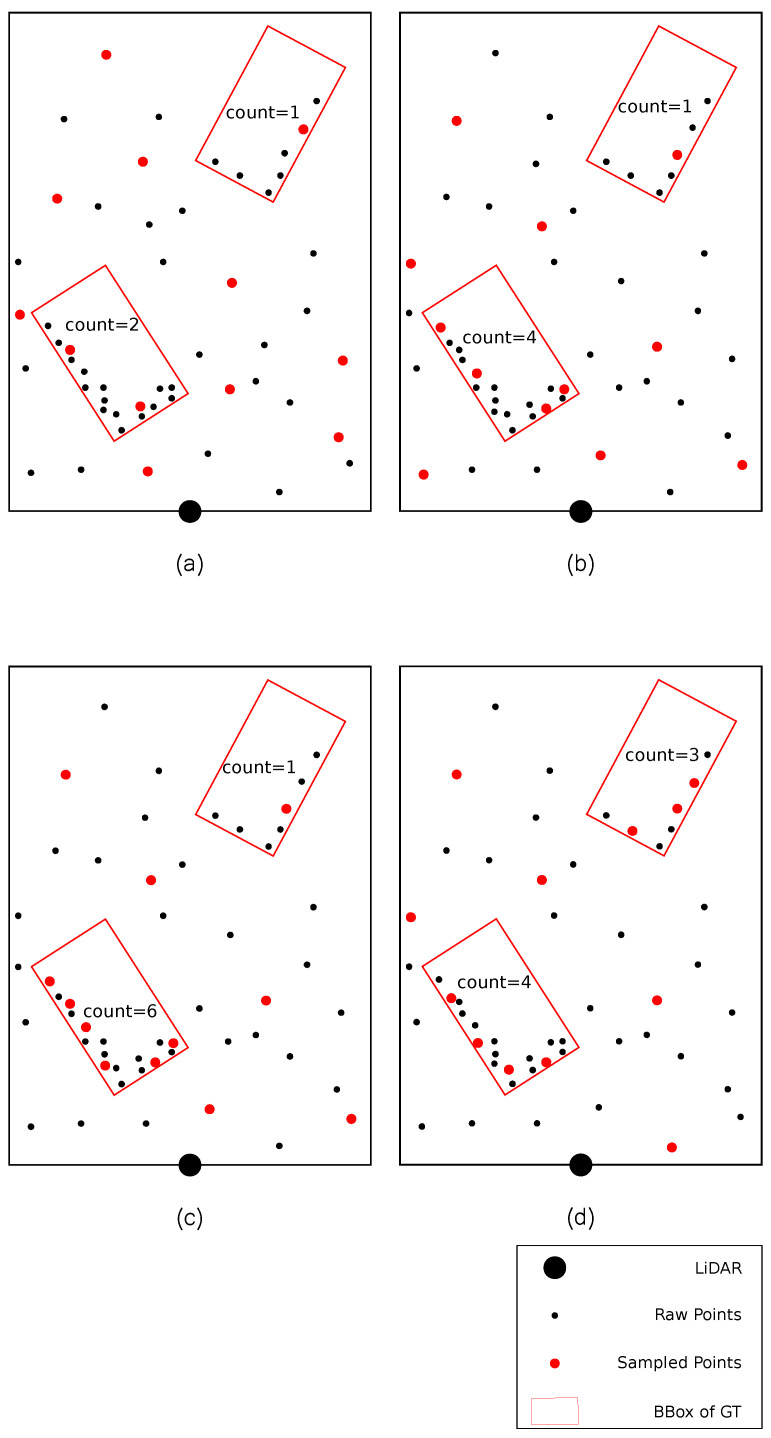
Points sampled by different sampling strategies. The sampled points are colored by red and enlarged for clarity. The black points in the bottom of each subdiagram denote the LiDAR position. The rectangles in each subdiagram denote the Bounding Box (BBox) of the GT. The number of sampled foreground points is annotated in each GT. (**a**) points sampled by vanilla FPS. The sampled points are evenly distributed in the sampling space (**b**) points sampled by F-FPS. F-FPS samples more foreground points than FPS. (**c**) points sampled by S-FPS. S-FPS samples more foreground points than F-FPS. (**d**) points sampled by DS-FPS. DS-FPS samples more foreground points than F-FPS and the sampling process is more balanced than S-FPS among instances with various density.

**Figure 2 sensors-23-05757-f002:**
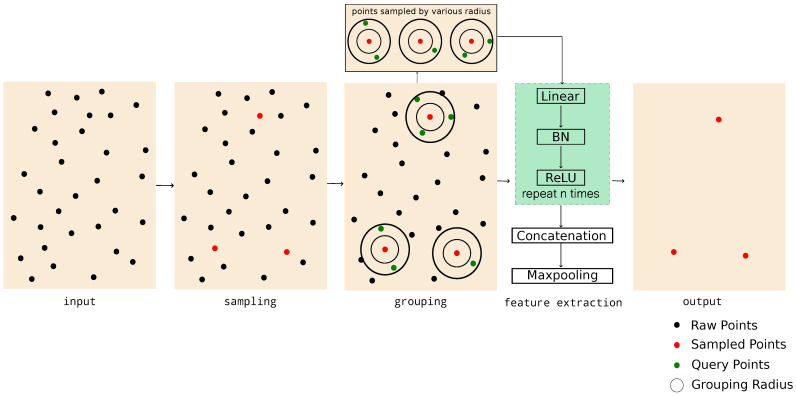
The overview of the SA module. The SA module first down-samples points; then, it uses different query radius to group points and feature extraction module repeats MLP (linear layer, BatchNorm (BN) layer and ReLU) for n times to better abstract the feature. A single MaxPooling layer is followed by the concatenation of multi-scale point features to obtain the final sampled points feature.

**Figure 3 sensors-23-05757-f003:**
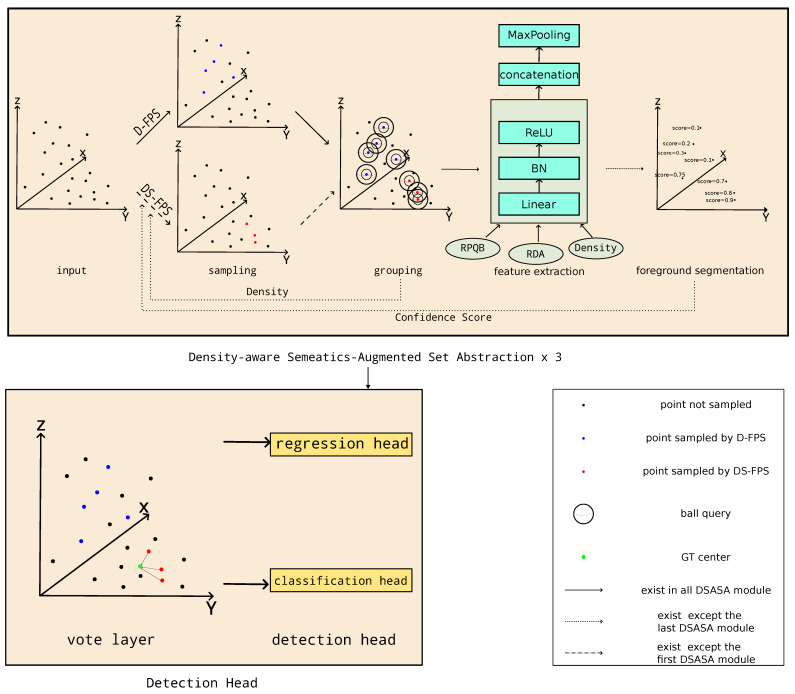
The framework of our DSASA. We repeat Density-aware Semantics-Augmented Set Abstraction three times. The first SA module only use FPS for sampling due to the inaccurate semantic feature in the early stage. The black points are points which are not sampled for the next stage. The blue points are sampled by D-FPS, the red points are sampled by DS-FPS, and the green point in the vote layer is the GT center which the points sampled by DS-FPS need to shift to. We feed the density and point level confidence score to DS-FPS to obtain a more balanced sampling distribution. The Relative Position in Query Ball (RPQB), Relative Direction Angle (RDA) and the point density are the extra input to the feature extraction module, which is detailed in Section 3.2.1.

**Figure 4 sensors-23-05757-f004:**
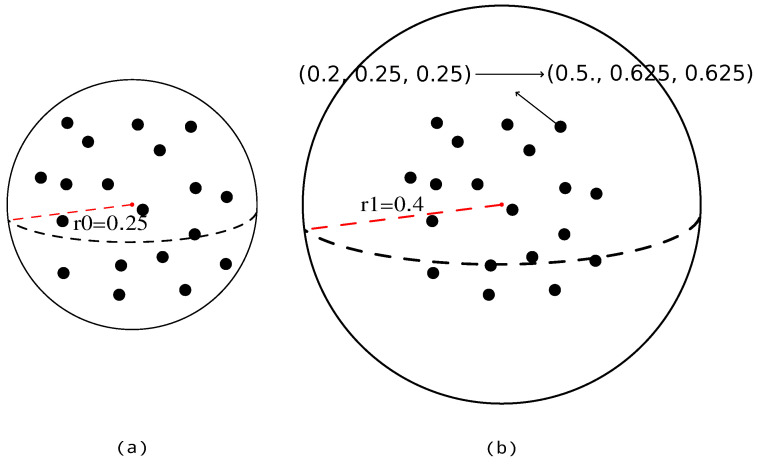
The importance of the distance to the query boundary. The red point is the center of the query ball. The black points are the queried points. The red dotted line means the radius of the query ball. The black dotted line means the circumference. (**a**) We use the fixed radius $r_{0}$ to query points, and the points are densely located in the ball. (**b**) We use a larger radius $r_{1}$ to query points, and the points are mainly located in the ball with radius $r_{0}$. They are two different circumstances, but in the vanilla SA module, it will generate the same feature. So, we normalize the relative position based on the radius, e.g., the relative coordinates (0.2, 0.25, 0.25) are converted to ((0.2−0)/(0.4−0), (0.25−0)/(0.4−0), (0.25−0)/(0.4−0)), that is (0.5, 0.625, 0.625).

**Figure 5 sensors-23-05757-f005:**
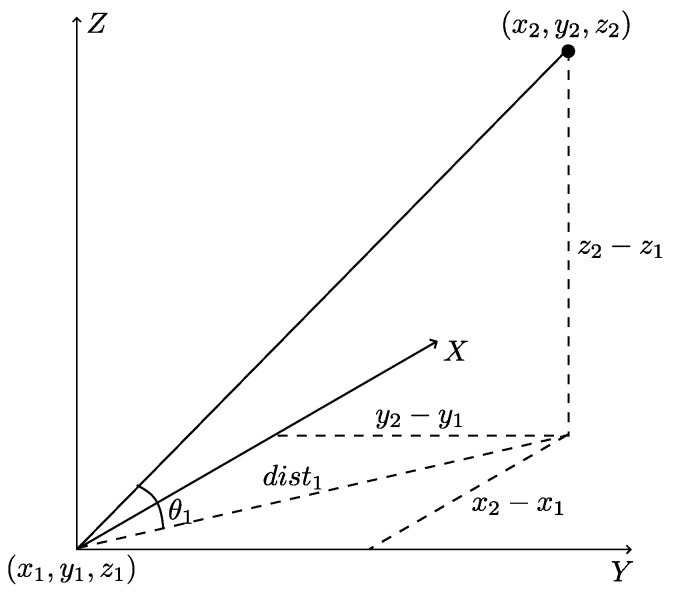
The illustration of relative direction angle. The original point is the center of the query ball. We can use the atan2 function to obtain $\theta_{1}$.

**Figure 6 sensors-23-05757-f006:**
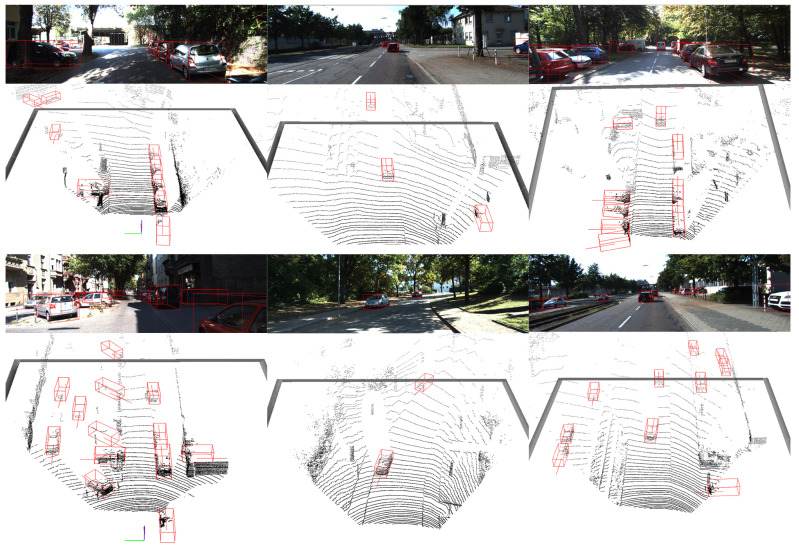
Results of 3D car detection on the KITTI test set. The predictions are labeled by red bounding boxes.

**Figure 7 sensors-23-05757-f007:**
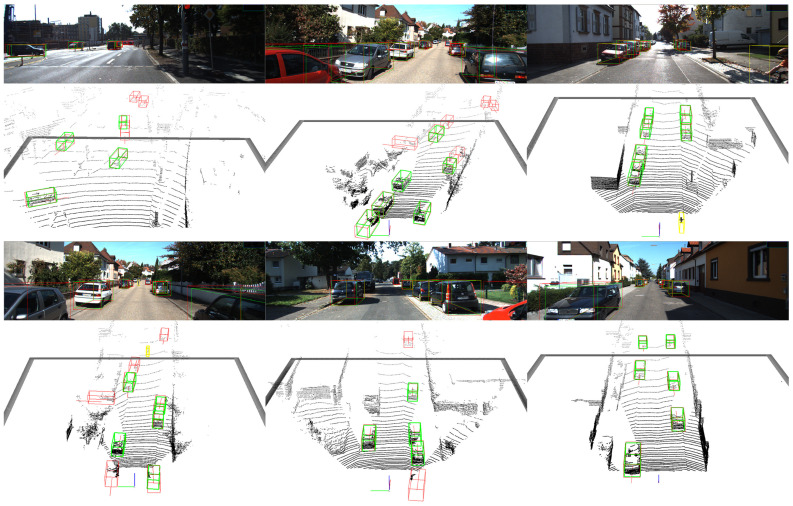
Results of 3D car detection on the KITTI validation set. The GTs are annotated by green bounding boxes and the predictions are labeled by red bounding boxes.

**Figure 8 sensors-23-05757-f008:**
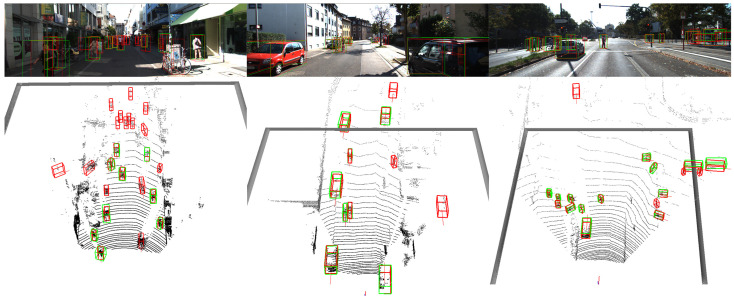
Results of multi-class 3D detection on the KITTI validation set. The GTs are annotated by green bounding boxes and the predictions are labeled by red bounding boxes.

**Figure 9 sensors-23-05757-f009:**
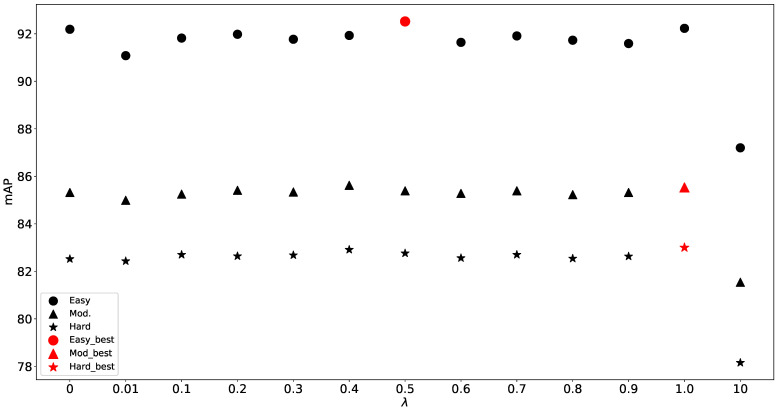
Performance with different balance factor λ.

**Table 1 sensors-23-05757-t001:** Results on the car class of the KITTI test set. The evaluation metric is the AP calculated on 40 recall points. The best results in each category are shown in **bold**.

Method	Car (IoU = 0.7)		
	Easy	Mod.	Hard
RGB+LiDAR			
MV3D [28]	74.97	63.63	54.00
F-PointNet [29]	82.19	69.79	60.59
Focals Conv-F [30]	90.55	82.28	77.59
VirConv-T [31]	**92.54**	**86.25**	**81.24**
LiDAR only			
**Voxel-based**			
VoxelNet [13]	77.47	65.11	57.73
SECOND [2]	83.34	72.55	65.82
PointPillars [14]	82.58	74.31	68.99
SA-SSD [32]	88.75	79.9	74.16
Voxel-RCNN [33]	**90.90**	81.62	77.06
VoxSeT [16]	88.53	**82.06**	**77.46**
**Hybrid-based**			
PV-RCNN [6]	**90.25**	81.43	76.82
PV-RCNN++ [7]	90.14	**81.88**	**77.15**
**Point-based**			
PointRCNN [8]	86.96	75.64	70.70
3DSSD [9]	88.36	79.57	74.55
IA-SSD [11]	**88.87**	80.32	75.10
SASA (reproduced) [10]	87.79	81.21	76.52
DSASA (ours)	88.64	**81.72**	**76.73**

**Table 2 sensors-23-05757-t002:** Results on the car class of the KITTI validation set. The evaluation metric is the AP calculated on 40 recall points. The best performance is shown in **bold**.

Method	Car (IoU = 0.7)			Delay (ms)
	Easy	Mod.	Hard	
PointRCNN	91.57	82.24	80.45	57
PointRCNN+SASA (reproduced)	92.14	83.10	80.71	48
PointRCNN+DSASA	92.05	84.05	82.55	50
3DSSD	91.54	83.46	82.18	**36**
3DSSD+SASA (reproduced)	91.89	85.32	82.52	**36**
3DSSD+DSASA	**92.54**	**85.91**	**83.12**	37

**Table 3 sensors-23-05757-t003:** Results on the 3 class of the KITTI validation set. The evaluation metric is the AP calculated on 40 recall points. The best performances are shown in **bold**.

Method	Car (IoU = 0.7)			Ped. (IoU = 0.5)			Cyc. (IoU = 0.5)		
	Easy	Mod.	Hard	Easy	Mod.	Hard	Easy	Mod.	Hard
PointRCNN	91.92	80.84	78.47	67.00	58.48	51.21	93.37	75.16	70.67
PointRCNN+SASA (reproduced)	92.13	82.76	80.39	68.34	60.48	51.92	92.30	74.13	69.71
PointRCNN+DSASA	92.25	82.93	80.60	**71.22**	**63.19**	**55.62**	**94.12**	**76.29**	**71.79**
3DSSD	91.47	83.00	81.88	57.10	52.24	48.83	89.90	71.78	68.09
3DSSD+SASA (reproduced)	92.02	85.32	82.55	63.28	57.98	53.45	92.20	74.37	69.74
3DSSD+DSASA	**92.18**	**85.32**	**82.71**	67.21	59.38	52.19	92.93	75.08	70.46

**Table 4 sensors-23-05757-t004:** Performance with different attributes encoded in RCE. The best performances are shown in **bold**.

Method	Car (IoU = 0.7)		
	Easy	Mod.	Hard
Baseline (DSASA without RCE)	92.23	85.53	83.00
Baseline+RPQB	92.13	85.57	82.92
Baseline+RDA	92.14	85.77	83.02
Baseline+Density	92.20	85.68	82.80
Baseline+ADA	91.54	85.65	82.83
Baseline+RPQB+RDA+Density	**92.54**	**85.91**	**83.12**
Baseline+RPQB+ADA+Density	91.72	85.69	82.77

**Table 5 sensors-23-05757-t005:** Performance with different attributes encoded in RCE. The best performances are shown in **bold**.

Method	Car (IoU = 0.7)		
	Easy	Mod.	Hard
RCE	**92.54**	**85.91**	**83.12**
Small MLP	91.64	85.66	82.97
Large MLP	91.89	85.47	82.61

**Table 6 sensors-23-05757-t006:** Mean and Std of different sampling methods.

	Stage	Mean	Std
F-FPS	Second SA	9.23	6.42
	Third SA	4.23	3.12
S-FPS	Second SA	23.31	20.89
	Third SA	20.37	18.29
DS-FPS	Second SA	17.89	15.00
	Third SA	16.67	13.84

## Data Availability

The dataset can be obtained on https://www.cvlibs.net/datasets/kitti/, accessed on 1 September 2022.

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
