# Peer review of "Boosting 3D Object Detection with Density-Aware Semantics-Augmented Set Abstraction"

_sensors, 2023, doi:10.3390/s23125757_

Round 1

Reviewer 1 Report

Incorporating density into point cloud analysis is super interesting and relevant. The proposal is interesting and well described. The testing, however, leaves a lot to be desired.

My main gripe with this paper is the lack of testing. The method is tested only on a single dataset, and thus it is impossible to say how well it generalizes - especially given the minor performance improvements shown. I would suggest also testing on a number of datasets significantly different to KITTI, e.g. S3DIS and ScanNet. Both of these have leaderboards online, making them easy to compare to. The author should also consider testing on the newer KITTI-360 dataset.

Minor comments:
- As far as references go, it seems strange to not include PointNeXt, as it is of the PointNet lineage already mentioned, and the best performing of those. There are also a wealth of transformer based methods which are glanced over in related work.
- On line 37: What does "obstacles" refer to?
- Fig. 6+7 are way too dark, impossible to read.
- Table 2: Why is evaluating on validation set relevant?

There are a number of grammar issues. On the first page e.g.

- "SA module can be split" -> "The SA module can be split" (line 4)

- "LiDAR sensors has been spreadly employed" -> LiDAR sensors have been broadly employed" (line 17)

- "in autonomous driving system" -> "in autonomous driving systems" (line 18).

Similar errors exist throughout the paper, though it is generally easily understandable.

Author Response

Details attached

Reviewer 2 Report

Well-written paper and proposes Density-aware semantics augmented set abstraction method.

The novelty of the work is clear.

Please relate the findings to previously reported studies.

The discussion of the research is clearly explained and described. But it would be better if the author(s) conveys the practical limitation and potential open problems for further research

Author Response

Details attached

Reviewer 3 Report

In this paper, the authors proposed density-aware based method to boost 3D object detection. The results show that the proposed method outperforms other methods in many aspects. However, I have following comments for improving the quality of this paper.

1) The writing style of abbreviations is not consistent throughout this paper. It is to use all abbreviations in the same way, such as using [Full Name (abbreviation)] at first use and just abbreviation in the following paragraphs. The paper should be carefully checked.

2) It is better to write Table 3 and 4 as the same style as Table 1 and 2. For example, The combination of the attributes in the 5th raw in Table 3 can be expressed as [PRQA+RDA+Density].

3) It is better to explain the reason why the attribute combinations of Table 3 are chosen.

In general it is well writen but some wrting style should be modified.

Author Response

Details attached

Round 2

Reviewer 1 Report

I appreciate the authors' changes and additions.

I can accept the argument that this work is main focused on self-driving and hence testing on indoor datasets is less relevant. I appreciate the addition of testing on multiple classes. That definitely enhances the paper.

Still, it it my opinion that testing the method on multiple datasets is a requirement. I don't understand the reluctance to test on KITTI-360. Yes, perhaps the baseline detectors will perform poorly, but we're merely interested in the relative difference between them and your approach (I note that there are results for both PointNet++ and PointNet for e.g. 3D semantic segmention, so it's certainly possible for them to run on KITTI-360). I also looked into the SASA paper, and they test not only on KITTI, but on nuScenes as well.

I feel my remaining comments have been addressed adequately.

Author Response

Details Attached
